# Role of Antimicrobial Resistance in Outcomes of Acute Endophthalmitis

**DOI:** 10.3390/antibiotics12081246

**Published:** 2023-07-28

**Authors:** Aaron Yap, Sharmini Muttaiyah, Sarah Welch, Rachael L. Niederer

**Affiliations:** 1Department of Ophthalmology, University of Auckland, Auckland 1142, New Zealand; aaron.yap8@gmail.com; 2Department of Ophthalmology, Te Whatu Ora Te Toka Tumai, Auckland 1051, New Zealand; swelch@adhb.govt.nz; 3Department of Microbiology, Te Whatu Ora Te Toka Tumai, Auckland 1023, New Zealand; sharminim@adhb.govt.nz

**Keywords:** antimicrobial resistance, visual outcome, endophthalmitis

## Abstract

Background: This study explores local trends in antimicrobial resistance and its influence on long-term visual outcomes following treatment with broad-spectrum empiric intravitreal antibiotics. Methods: All patients undergoing intraocular sampling for endophthalmitis from Auckland between January 2006–May 2023 were included. The impact of antimicrobial resistance on the final visual outcome was analysed using logistic regression models. Results: 389 cases of endophthalmitis were included, and 207 eyes (53.2%) were culture positive. When tested, all Gram-positive microorganisms were fully susceptible to Vancomycin, and all Gram-negative microorganisms demonstrated full or intermediate susceptibility to Ceftazidime. Resistance to at least one antimicrobial agent was present in 89 culture results (43.0%), and multidrug resistance (resistant to ≥3 antimicrobials) in 23 results (11.1%). No increase in resistance was observed over time. The primary procedure was a tap and inject in 251 eyes (64.5%), and early vitrectomy was performed in 196 eyes (50.3%). Severe vision loss (≤20/200) occurred in 167 eyes (42.9%). Antimicrobial resistance was associated with an increased risk of retinal detachment (OR 2.455 *p* = 0.048) but not vision loss (*p* = 0.288). Conclusion: High sensitivity to Vancomycin and Ceftazidime was present in our population, reinforcing their role as first-line empiric treatments. Resistant microorganisms were associated with an increased risk of retinal detachment but no alteration in final visual outcome.

## 1. Introduction

Antimicrobial therapy is one of the most important medical discoveries of the twentieth century, and it has prevented millions of cases of premature morbidity and mortality from bacterial infection. Since antimicrobials have become widely available, the burden of resistance among bacteria has increased in tandem, especially within the past 10 years. In endophthalmitis, where rapid control of infection is important for maintaining sight, reported rates of antimicrobial resistance have risen in northern American states [1,2]. This finding mandates an updated review of local trends in antimicrobial resistance and its impact on long-term visual function.

Local patterns of antimicrobial resistance are a major consideration in the choice of empiric antimicrobial therapy for endophthalmitis. Empiric intravitreal administration of antimicrobials, such as Vancomycin plus either Amikacin or Ceftazidime, is widely adopted, as they cover Gram-positive and Gram-negative microorganisms, respectively [3,4,5]. In India, there have been few case reports of emerging antimicrobial resistance against these antibiotics [6,7,8,9]. Historical reports from Auckland (1983–1991) and Queensland, Australia (1998–2013) have provided assurance that this trend has not spread to the Australasian region [10,11]. 

Microorganisms evolve to develop antimicrobial resistance through two mechanisms: (1) mutations in genes associated with drug target sites and (2) horizontal gene transfer of foreign DNA containing resistance determinants [12]. When used, antibiotics exert selection pressure for microorganisms possessing resistance mechanisms, which now have a survival advantage and are able to transfer forward resistance genes. This relationship is demonstrated in the hospital environment, where antimicrobial use is associated with the emergence of multiresistant strains [13,14]. From a broader perspective, trends in antimicrobial resistance relate to the inappropriate use of antibiotics in agriculture, community infections, healthcare policy, infection control and host migration [15,16,17].

Interpretation of antimicrobial resistance in clinical practice is a relative phenomenon with many layers of complexity. The establishment of clinical susceptibility breakpoints relies on the in vitro activity of an antibiotic against a sizeable bacterial sample, but it does not take into consideration the pharmacodynamics and pharmacokinetics of the drug, which vary according to the route and site of antibiotic administration. In vitro resistance does not necessarily equate to treatment failure in the context of intraocular pathogens, as high intraocular concentrations of antibiotics following an intravitreal bolus dose in the early stages of infection can overwhelm mechanisms of resistance [18]. Wu et al., found that antimicrobial resistance conferred no additional risk of vision loss provided all patients received empiric intravitreal antibiotics [5].

This study aims to describe the implicated microorganisms in endophthalmitis, their antimicrobial susceptibility profile, and clinical outcomes in a New Zealand population following a standardized treatment protocol that incorporates empiric treatment with intravitreal Vancomycin and Ceftazidime. 

## 2. Materials and Methods

### 2.1. Subject Selection

Subjects were recruited from a combined catchment area of 1.8 million residents of both Auckland and Northland, New Zealand [19]. All aqueous and vitreous samples received by the laboratory with a presumptive diagnosis of endophthalmitis between 1 January 2006 and 31 May 2023 were included in this study. Clinical note review was undertaken to supplement available laboratory data. In subjects with bilateral endogenous endophthalmitis, only the worst affected eye at presentation was included.

### 2.2. Data Collection

Data collection followed a standardised pro forma, which included subject demographics, clinical presentation, laboratory findings, complications of endophthalmitis and visual acuity at 3, 6, 9, 12 months and final follow-up. Final follow-up was defined as the most recent visual acuity (VA) recorded. 

Snellen best-corrected visual acuity (BCVA) was converted to logarithm of the minimum angle of resolution (LogMAR) for the purposes of analysis. The following conversion factors were used for BCVA of counting fingers (CF) or worse: CF, 2.0 LogMAR; hand motions, 2.3 LogMAR; light perception, 2.6 LogMAR; no light perception, 2.9 LogMAR [20]. Severe visual loss (SVL) was defined according to the Standardisation of Uveitis Nomenclature criteria as a permanent reduction in BCVA to ≤6/60 [21].

The primary outcome measure was SVL at final follow-up. Secondary outcomes were the rates of retinal detachment, enucleation or evisceration. 

### 2.3. Intraocular Sampling and Treatment

Intraocular specimens were applied onto a glass slide, and a Gram stain was performed to guide empiric treatment. The samples were then inoculated on appropriate media, which were sheep blood agar, GC Saponin agar, Sabouraud dextrose agar, brain heart infusion (BHI) agar and BHI broth for bacterial and fungal cultures. All media were incubated in 5% carbon dioxide at 37 °C except for the BHI agar, which was incubated at 37 °C anaerobically, and the Sabouraud dextrose agar, which was incubated at 27 °C aerobically. The agar plates were examined for growth at 24 and 48 h. The BHI broth was held for extended incubation for six days, and the BHI agar for ten days. If the broth became cloudy, it was subcultured onto sheep blood agar, chocolate agar and BHI agar and re-incubated for another 48 h. Since 2012, any bacterial colonies on the plates were identified by matrix-assisted laser desorption ionization-time of flight (MALDI-TOF) mass spectrometry using Vitek MS (BioMérieux, Marcy-l’Étoile, France) [22]. The current MS Knowledge database in use is version 3.4.

Between 2006 and 2016, antibiotic susceptibility testing was based upon Clinical Laboratory Standards Institute (CLSI, Wayne, PA, USA) methodology. From 2016 onwards, there was a transition from CLSI to the European Committee on Antimicrobial Susceptibility Testing (EUCAST) based on antimicrobial susceptibility breakpoints [23]. Although in 2019, EUCAST recommended a shift from the “intermediate” to “susceptible, increased exposure” criteria, this was not formally instated until 2020 due to the coronavirus pandemic [24].

Anterior chamber and vitreous tap samples were subjected to a panel of antimicrobial susceptibility testing (Appenndix Table A1) with either an automated method using Vitek 2 (bioMe’rieux, Marcy-l’Étoile, France) or the Kirby–Bauer (KB) disk diffusion method [25]. Vitek 2 cards were automatically filled, sealed, and loaded into the VITEK 2 instrument for incubation and reading. The Vitek 2 system detects growing bacteria on the basis of turbidity and is incorporated with advanced expert system software that provides the minimum inhibitory concentration (MIC) for each antibiotic. The Kirby–Bauer disk diffusion method is used for antimicrobials not included in the VITEK 2 system or for fastidious microorganisms that do not grow in the VITEK 2 system media, such as all *Streptococcus* spp., *Corynebacterium* spp., *Haemophilus* spp. and *Moraxella* spp. The KB disk diffusion method was performed using a standardised inoculum on a solid agar plate, and the antibiotic-treated disks were stamped on the inoculated plate. The disk containing the antibiotic was allowed to diffuse through the solidified agar, resulting in the formation of an inhibition zone after overnight incubation at 35 °C aerobically. After 16 to 24 h, the inhibition zone diameter was measured. The medium used for the Kirby–Bauer disk method was Mueller–Hinton or Mueller–Hinton Fastidious (MH-F) (Fort Richard Laboratories, Auckland, New Zealand).

All subjects received intravitreal Ceftazidime 2.25 mg in 0.1 mL and Vancomycin 1 mg in 0.1 mL following the procurement of an intraocular sample and at the conclusion of vitrectomy surgery. A standard preparatory solution of Povidine-Iodine 0.5% or Chlorhexidine 0.1% was used prior to sampling. A pure vitreous sample was performed at the start of vitrectomy using the controlled air-aspirate vitrectomy technique [26]. Non-surgical vitreous samples were obtained using a 25-gauge needle and syringe.

### 2.4. Statistical Analysis

All data were entered into an Excel spreadsheet and analysed in STATA version 15. Continuous data are presented as mean ± standard error of the mean (SEM), while categorical data are presented as median and interquartile range (IQR) or n (%). Multivariate analysis and relative risks for drug resistance, retinal detachment, enucleation and severe vision loss were performed using logistic regression modelling. Tests for normality were performed with the Shapiro–Wilk test. Visual acuity at follow-up was right-skewed and therefore was reported as median with interquartile range as well as mean to best describe the data. Variables with *p* < 0.150 on the univariate analysis were included in the multivariate analysis. A *p* value of ≤0.05 was considered statistically significant. 

## 3. Results

Three hundred eighty-nine cases of endophthalmitis were included during the study period. The median age at presentation was 70.0 years [IQR 58.1–80.0], and 203 (52.2%) were female. The underlying cause of endophthalmitis was cataract surgery in 117 (30.1%), intravitreal anti-vascular endothelial growth factor (VEGF) injection in 97 (24.9%), endogenous in 57 (14.7%), post vitrectomy in 32 (8.2%), post glaucoma surgery in 29 (7.5%), following corneal infection or surgery in 5 (1.3%), and following trauma or other procedures in 52 (13.4%).

The median duration of symptoms at presentation was one day [IQR 1–3 days]. The median time following surgery to presentation was 7.5 days [IQR 3–42] for cataract surgery, 4 days [IQR 2–6.5] for intravitreal injection, 1206 days [IQR 38–2746] for glaucoma surgery, 3 days [IQR 2–10] for post vitrectomy, and 3 days [IQR 2–16 days] for corneal surgery. The median time to presentation following trauma was 5 days [IQR 2–8]. The mean presenting BCVA was 1.88 ± 0.80 LogMar. The median BCVA was hand movement [IQR 20/400–hand movement]. Hypopyon was present in 202 (51.9%), and red reflex was present in 147 (37.8%) and absent in 242 (62.2%). 

A positive culture was obtained in 207 eyes (53.2%). An anterior chamber tap was performed in 167 eyes (42.9%) and was positive in 50 (29.9%). A vitreous tap was performed in 228 (58.6%) and was dry in 38. Of the 190 successful vitreous taps, culture results were positive in 83 (43.7%). Vitrectomy was performed in 247 eyes (63.5%) and was culture positive in 105 (42.5%). If vitrectomy was the primary procedure, it was culture positive in 62 of 122 (50.8%), whereas if vitreous tap and inject was the primary procedure, vitrectomy was positive in 42 of 138 (30.4%).

Organisms were Gram-positive in 162 eyes (79.8%), Gram-negative in 29 eyes (14.2%) and fungal in 12 eyes (5.9%) (Table 1). All Gram-positive microorganisms were tested for susceptibility to Vancomycin. In the nine Gram-negative microorganisms tested for Ceftazidime susceptibility, only one (11.1%) demonstrated intermediate susceptibility. Resistance to at least one antimicrobial agent was present in 89 culture results (43.0%), and multidrug resistance (resistance to ≥3 drugs) was present in 23 culture results (11.1%). In terms of fluoroquinolone resistance, two samples (3.2%) were resistant to ciprofloxacin, and all five (0%) microorganisms tested against moxifloxacin were susceptible. Resistance to at least one antimicrobial agent and multidrug resistance for *Staphylococcus aureus* was 63.2% and 5.3%, respectively. Similarly, for coagulase-negative staphylococci (CoNS), it was 69.1% and 17.6%, respectively.

There were no predictors for any resistance or multidrug resistance observed in patient demographics (age, gender) or in clinical presentation (cause of endophthalmitis, presenting vision, hypopyon, red reflex). No increase in the likelihood of resistant microorganisms was observed over time (OR 0.974 *p* = 0.345), and no association was observed between year of presentation and multidrug resistance (OR 1.037 *p* = 0.419). The frequency of resistance and multidrug resistance by year are reported in Table 2. 

The primary procedure was a tap and inject in 251 eyes (64.5%) and a vitrectomy in 138 eyes (35.5%). In those receiving a primary tap and inject, a further 58 underwent a secondary vitrectomy <24 h from presentation. Late vitrectomy (>24 h) occurred in 33 eyes (28.7%) with no antimicrobial resistance and in 19 eyes (21.8%) with at least one antimicrobial resistance (*p* = 0.270). There was also no association observed between late vitrectomy and multidrug resistance (25.6% with no multidrug resistance vs. 27.3% with multidrug resistance, *p* = 0.862). 

Median follow-up time was 10.1 months [IQR 2.3–34.1]. Retinal detachment occurred in 44 eyes (11.3%), and enucleation/evisceration in 29 eyes (7.5%). Mean visual acuity was 1.13 ± 1.03 LogMar at three months, 1.15 ± 1.07 LogMar at six months, and 1.16 ± 1.10 LogMar at nine months. Median visual acuity was 20/80 [IQR 20/40–hand movement] at three months, 20/100 [IQR 20/30–hand movement] at six months, and 20/100 [IQR 20/30–hand movement] at nine months. At the last recorded follow-up, mean visual acuity was 1.18 ± 1.06 LogMar, and median visual acuity was 20/100 [IQR 20/30–hand movement]. Severe vision loss (≤20/200) occurred in 167 eyes (42.9%). 

Risk factors for retinal detachment are reported in Table 3. On univariate analysis, the following variables were associated with retinal detachment: younger age (OR 0.971 *p* < 0.001); weekend presentation (OR 2.151 *p* = 0.025); presenting visual acuity (OR2.977 *p* = 0.002); hypopyon (OR 2.274 *p* = 0.028); and absence of red reflex (OR 0.400 *p* = 0.014). On multivariate analysis, the following variables were associated: younger age (OR 0.977 *p* = 0.028); hypopyon (OR 3.239 *p* = 0.048); and antimicrobial resistance (OR 2.455 *p* = 0.048). 

Risk factors for enucleation or evisceration are reported in Table 4. On univariate analysis, only presenting visual acuity was associated with increased risk of enucleation or evisceration (OR 2.965 *p* = 0.013). On multivariate analysis, there were no associated risk factors observed. No significant association was observed between antimicrobial resistance and risk of enucleation. 

Risk factors for severe vision loss (≤20/200) are reported in Table 5. On univariate analysis, the following were associated with increased risk of vision loss: younger age (OR 0.989 *p* = 0.042); poor presenting vision (OR 3.451 *p* < 0.001); and absence of red reflex (OR 0.362 *p* < 0.001). On multivariate analysis, poor presenting vision was associated with increased risk of vision loss (OR 3.323 *p* < 0.001), and early vitrectomy (<24 h) was associated with reduced risk (OR 0.565 *p* = 0.020). There was no association between antimicrobial resistance or multidrug resistance with severe visual acuity loss. 

## 4. Discussion

In light of increasing antimicrobial resistance globally, this study describes trends in antimicrobial resistance in New Zealand and its effect on patient outcomes. Vancomycin and Ceftazidime provided comprehensive coverage over Gram-positive and Gram-negative organisms, respectively. Resistance to at least one organism was present in nearly half of the samples, and multidrug resistance in approximately 10%. There was no trend towards increasing antimicrobial resistance over time. Whilst antimicrobial resistance was not associated with an increased risk of vision loss or enucleation, it was associated with increased risk of retinal detachment.

Empiric treatment plays a crucial role in limiting vision loss by halting microbial proliferation and should not be delayed. In the maxim of “Time is Retina”, Michael et al., retrospectively reviewed 374 eyes with endophthalmitis and found that treatment within 2 h with intravitreal antibiotics was associated with a better final visual outcome [27]. Historically, dual therapy, with Vancomycin directed against Gram-positive organisms and either Amikacin or Ceftazidime against Gram-negative organisms, has been established as a rational choice to cover a heterogenous group of causative organisms [3,28]. Whilst this combination of antibiotics is validated by low resistance rates in our study population, the same does not apply to other centres that report emerging cases of antimicrobial resistance [6,7,8,9,29]. In comparison with the ARMOR surveillance study, there was at least a two-fold rate increase in multidrug resistance in CoNS (46.3%) and *Staphylococcus aureus* (41%) isolates [30]. Such resistant organisms are associated with poor visual outcomes and warrant regular reviews of local empiric antibiotic selection.

Antimicrobial stewardship constitutes coordinated interventions designed to improve and measure the appropriate use of antimicrobials by promoting the selection of the optimal antimicrobial drug regimen, dose, duration of therapy, and route of administration [31]. New Zealand fares better than other nations, with comparatively lower rates of antimicrobial resistance and conservative antibiotic usage, and this acts as a strong motivator for collaborative efforts to stop the emergence of further resistance. Nationwide initiatives include: (1) antimicrobial prescribing guidelines enabling prescribers to select an effective agent at the correct dose with the narrowest spectrum, fewest adverse effects and lowest cost [32], (2) continuous surveillance of all clinical laboratories’ susceptibility testing for emerging antibiotic resistance, (3) infection control policy that controls the spread of resistant strains and decreases overall use of antimicrobials [17,33], (4) all antibiotics supplied by prescription only, (5) infectious disease approval is required for last-line antibiotics, and (6) antibiotics in farming animals are regulated for therapeutic and prophylactic purposes, as opposed to large-scale use as growth promoters in other parts of the world [16,31,34].

Local protocols instituted in the studied eye department include: (1) avoiding the use of topical antimicrobials before or after intravitreal injections, (2) utilising intracameral cefuroxime for infection prophylaxis following routine cataract surgery instead of topical antimicrobials, and (3) judicious use of topical or oral antimicrobials for ocular infections. These local measures may decrease the presence of resistant organisms within a single individual’s ocular flora but do not make much difference in levels of resistance within the community. There is good evidence that topical antimicrobials do not minimise the risk of endophthalmitis following intravitreal injection [35]. On the contrary, prolonged use of antimicrobials eliminates the natural flora and favours proliferation and potential infection with antimicrobial-resistant organisms [36,37]. Intracameral cefuroxime at the conclusion of routine cataract surgery provides the best protection against post-operative endophthalmitis, with topical antimicrobials conferring little added benefit [38,39,40]. Finally, the use of fluoroquinolones, such as topical ciprofloxacin and oral moxifloxacin, is strictly regulated. Prescribing rights are exclusive to tertiary ophthalmology service providers, and oral moxifloxacin is reserved for use in penetrating eye trauma. 

Our cohort of culture-positive endophthalmitis (n = 207) is the largest to date that addresses conflicting views about the impact of antimicrobial resistance on visual outcomes. Whilst Wu et al., (n = 99) found that antimicrobial resistance was not associated with a risk of vision loss, Choi et al., (n = 82) found that resistance to Vancomycin or third-generation cephalosporins was associated with a 75% lower chance of achieving VA better than counting fingers [5,41]. Further complicating matters is the lack of concordance between in vitro antimicrobial sensitivities and clinical response, which is reflective of greater concentrations of antimicrobials following intravitreal administration. Following an intraocular injection of Vancomycin at the same treatment dose for endophthalmitis, concentrations exceeded the minimum inhibitory concentration (MIC) of endophthalmitis-causing Gram-positive bacteria four-fold for up to 26 h. [18] Alternative measurements, such as the minimal bactericidal concentration or time-kill curve of antibiotics, may provide greater value in infections in immune-privileged sites such as the inner chamber of the eye [42].

The association between antimicrobial resistance and higher rates of retinal detachment would suggest underlying involvement of other virulence factors. In a rabbit model of *Staphylococcus epidermidis*-induced endophthalmitis, Kaspar et al., found that antimicrobial-resistant strains caused more inflammation and destruction of the infected retina than antimicrobial-susceptible strains. One plausible explanation is that virulence factors are genetically transferred together with resistance vectors, accumulating in the longer-surviving bacteria [43]. These bacteria are better capable of ocular tissue invasion, surviving in the intraocular compartment, breaking down the blood–retinal barrier and triggering a destructive immune response in the retina [44]. The fact that weekend presentations are also associated with retinal detachments could be reflective of either greater symptom severity or longer delay to initial presentation. Previous work by our research group has not observed any delay in treatment with weekend presentations nor a decreased likelihood of receiving a vitrectomy over the weekend [27]. 

Visual outcome based upon the proportion of patients without severe vision loss is better than previously reported, which could be attributable to differences in study population, research protocols and treatment regimens. Wu et al., conducted a study utilising a similar protocol and found a higher rate of severe vision loss compared to our cohort (42.9% vs. 67.7%) and a worse mean final BCVA (2.19 LogMar vs. 1.18 LogMar) [5]. As presenting vision and culture positivity rates were equivocal, other possible explanations for this difference could be the higher degree of reported antimicrobial resistance and lower rates of early vitrectomy (50.4% vs. 4.0%). Early vitrectomy, in particular, has been shown to be predictive of better long-term visual outcome [45]. 

There was a steep rise in both incidence rates and percentage of culture-positive cases from 2019 to 2021, coinciding with the start of the COVID-19 pandemic. Whilst the number of elective cataract and vitreoretinal surgeries decreased, intravitreal injections continued largely unabated during the COVID pandemic. Mandatory mask wearing, social distancing and lockdowns may also have played a role in the disease spectrum. Finally, the year-to-year variation in the percentage of culture-positive cases may be associated with the likelihood of diagnosing endophthalmitis versus sterile inflammation, as all those managed with intravitreal antibiotics and sampling were included in our study cohort.

The limitations of this study are inherent to retrospective analysis, which include incomplete data, selection bias and lack of randomisation. However, it is the largest cohort to date with comprehensive long-term data allowing for visual prognostication based on baseline factors and treatment. Although our protocol for antimicrobial susceptibility follows global standards, subtle variations in antimicrobial selection represent a potential source of sampling bias. Lower quantities of antimicrobials selected for testing result in underdetection of antimicrobial resistance, and vice versa. In line with local antimicrobial stewardship initiatives, first-line antibiotics assume priority in testing and reporting to promote their clinical use. 

## 5. Conclusions

Rates of antimicrobial resistance in endophthalmitis are lower in the Auckland region, which may be explained by the implementation of judicious antimicrobial stewardship. In conjunction with rapid intravitreal administration of broad-spectrum antibiotics and early vitrectomy, long-term visual outcomes are better than previously reported. This study shows that, in our cohort, antimicrobial resistance may increase the likelihood of retinal detachment but does not appear to significantly alter final visual outcome or the likelihood of further procedures. 

## Figures and Tables

**Table 1 antibiotics-12-01246-t001:** List of microorganisms in culture-positive endophthalmitis.

Microorganism	Count
**Gram-positive**	
*Abiotrophia defectiva*	1
*Aerobic sporing bacillus*	2
*Clostridium perfringens*	1
*Coagulase-negative staphylococcus*	4
*Corynebacterium Group G*	1
*Corynebacterium jeikeium*	1
*Enterococcus faecalis*	9
*Enterococcus faecium*	1
*Enterococcus hirae*	1
*Gram-positive coccobacilli*	1
*Granulicatella adiacens*	2
*Lactococcus*	1
*Moraxella species*	1
*Methicillin-resistant Staphylococcus aureus*	4
*Mycobacterium chelonae*	1
*Propionibacterium acnes*	1
*Rothia dentocariosa*	1
*Staphylococcus aureus*	16
*Staphylococcus epidermidis*	64
*Staphylococcus lugdunensis*	4
*Streptococcus agalactiae*	1
*Streptococcus dysgalactiae*	1
*Streptococcus Lancefield Group C*	2
*Streptococcus Lancefield Group G*	2
*Streptococcus mitis*	11
*Streptococcus oralis*	5
*Streptococcus parasanguinis*	1
*Streptococcus pneumoniae*	6
*Streptococcus pyogenes*	4
*Streptococcus salivarius*	3
*Streptococcus sanguinis*	6
*Streptococcus viridans*	3
**Total**	**162**
**Gram-negative**	
*Aggregatibacter actinomycetemcomitans*	2
*Escherichia coli*	2
*Haemophilus influenzae*	8
*Klebsiella pneumoniae*	5
*Morganella morganii*	1
*Pseudomonas aeruginosa*	7
*Serratia marcescens*	3
*Stenotrophamonas maltophilia*	1
**Total**	**29**
**Fungal**	
*Alternaria species*	1
*Aspergillus fumigatus*	2
*Candida albicans*	3
*Candida parapsilosis*	3
*Candida rugosa*	1
*Fusarium solani*	1
*Scedosporium apiospermum*	1
**Total**	**12**

**Table 2 antibiotics-12-01246-t002:** Rates of antimicrobial resistance over time.

Year	Total Cases (n = 389)	Culture Positive	Any Resistance ^1^	Multidrug Resistance ^1^
2006	16	9	3 (33.3%)	0 (0%)
2007	22	13	5 (38.5%)	1 (7.7%)
2008	16	11	2 (18.2%)	1 (9.1%)
2009	12	10	7 (70.0%)	0 (0%)
2010	14	5	3 (60.0%)	1 (20.0%)
2011	28	9	3 (33.3%)	1 (11.1%)
2012	17	8	5 (62.5%)	1 (12.5%)
2013	20	9	6 (66.7%)	1 (11.1%)
2014	24	16	9 (56.3%)	1 (6.3%)
2015	20	7	4 (57.1%)	0 (0%)
2016	27	8	4 (50.0%)	2 (25.0%)
2017	29	19	7 (36.8%)	4 (21.1%)
2018	25	12	7 (58.3%)	2 (16.7%)
2019	41	18	7 (38.9%)	1 (5.6%)
2020	25	14	6 (42.9%)	3 (21.4%)
2021	30	23	6 (26.1%)	3 (13.0%)
2022	14	6	3 (50.0%)	0 (0%)
2023	11	10	2 (20.0%)	0 (0%)

^1^ Percentages expressed as percentage of culture-positive cases; 2023 data represent only part of a year to May 2023.

**Table 3 antibiotics-12-01246-t003:** Predictors of retinal detachment.

Risk Factor	Univariate		Multivariate	
	Odds Ratio	*p*-Value	Odds Ratio	*p*-Value
Age	0.971	<0.001	0.977	0.028
Female	0.587	0.101	0.800	0.637
Year of presentation	0.986	0.687		
Weekend presentation	2.151	0.025	2.587	0.058
Presenting vision	2.977	0.002	1.849	0.179
Hypopyon	2.274	0.028	3.239	0.048
Red reflex	0.400	0.014	1.921	0.258
Early vitrectomy	1.159	0.659		
Any resistance	1.888	0.104	2.455	0.048
Multidrug resistance	1.765	0.308		

**Table 4 antibiotics-12-01246-t004:** Predictors of enucleation or evisceration.

Risk Factor	Univariate		Multivariate	
	Odds Ratio	*p*-Value	Odds Ratio	*p*-Value
Age	0.986	0.108	1.000	0.978
Female	0.535	0.115	0.558	0.258
Year of presentation	0.907	0.017	0.922	0.108
Weekend presentation	0.883	0.794		
Presenting vision	2.965	0.013	1.527	0.405
Hypopyon	1.531	0.319		
Red reflex	0.305	0.019	0.821	0.766
Early vitrectomy	0.677	0.369		
Any resistance	0.952	0.912		
Multidrug resistance	2.684	0.084	2.733	0.124

**Table 5 antibiotics-12-01246-t005:** Predictors of severe vision loss (BCVA ≤ 20/200).

Risk Factor	Univariate		Multivariate	
	Odds Ratio	*p*-Value	Odds Ratio	*p*-Value
Age	0.989	0.042	0.993	0.286
Female	0.828	0.354		
Year of presentation	0.972	0.185		
Weekend presentation	0.832	0.462		
Presenting vision	3.451	<0.001	3.323	<0.001
Hypopyon	1.042	0.848		
Red reflex	0.362	<0.001	0.900	0.717
Early vitrectomy	0.729	0.142	0.565	0.020
Any resistance	0.739	0.288		
Multidrug resistance	0.915	0.844		

## Data Availability

The data presented in this study are available on request from the corresponding author. The data are not publicly available to protect patient privacy.

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
