# Peer review of "Role of Antimicrobial Resistance in Outcomes of Acute Endophthalmitis"

_antibiotics, 2023, doi:10.3390/antibiotics12081246_

Round 1

Reviewer 1 Report

The authors describe trends of antimicrobial resistance in New Zealand and its effect on patient outcomes. However, there are some questions need to be addressed. 

1. In ‘Introduction’, it would be better to add some reasons why ‘increasing microbial resistance against a variety of antimicrobials has been observed’.

2. In ‘Introduction’, I think this part is very short and it would be better to add more background and more references in this field.

3. In ‘Methods’, please indicate which statistical methods used in this research.

4. In ‘Results’, it would be better to represent data as mean and SD/SEM.

5. In ‘Discussion’, there are too much description in this part especially compared with the ‘Introduction’.

Reviewer 2 Report

In the file review 2502633

Reviewer 3 Report

AMR is a global issue that is driven the antimicrobial reagent studies. Instead of looking for new medicines, finding a sensitive treatment from the current drug list is more important. The authors presented the antimicrobial resistance of eye infections in New Zealand from 2006 to 2023, the data not only showed the trend of AMR in New Zealand, but also indicated vancomycin and ceftazidime have positive effects in clinical treatment. The data and conclusions are reasonable and impressive. I think this paper can provide a good guideline for clinical treatment. I just have one suggestion, in the table 1 from the year of 2019 to 2021, the sample size increased but the positive culture ratio decreased. Could it be associated with the pandemic policy or virus treatment? Just a guess, and it may reveal more risk factors.

Round 2

Reviewer 2 Report

No comments

Author Response

Dear Reviewer,

Thank you for reviewing my article.

I appreciate your expertise in the field and the effort gone in to critically appraise our work.